# Gene Expression Profiles Deciphering the Pathways of Coronatine Alleviating Water Stress in Rice (*Oryza sativa* L.) Cultivar Nipponbare (Japonica)

**DOI:** 10.3390/ijms20102543

**Published:** 2019-05-23

**Authors:** Wei Gao, Chunxin Yu, Lin Ai, Yuyi Zhou, Liusheng Duan

**Affiliations:** Engineering Center for Plant Growth Regulators MOE, College of Agronomy and Biotechnology, China Agricultural University, Beijing 100193, China; wei1227@126.com (W.G.); yuchunxin@cau.edu.cn (C.Y.); 18601272095@163.com (L.A.); zhouyuyi711@126.com (Y.Z.)

**Keywords:** drought stress, gene expression, coronatine, phytohormone

## Abstract

Coronatine (COR) is a structural and functional analog of methyl jasmonic acid (MeJA), which can alleviate stress on plant. We studied the effects of COR on the drought stress of rice (*Oryza sativa* L.). Pre-treatment with COR significantly increased the biomass, relative water and proline content, and DPPH (1,1-diphenyl-2-picrylhydrazyl)-radical scavenging activity, decreased the electrolyte leakage and MDA (Malondialdehyde) content in order to maintain the stability of cell membrane. Meanwhile, we determined how COR alleviates water stress by Nipponbare gene expression profiles and cDNA microarray analyses. Seedlings were treated with 0.1 μmol L^−1^ COR at the three leafed stage for 12 h, followed with 17.5% polyethylene glycol (PEG). Whole genome transcript analysis was determined by employing the Rice Gene Chip (Affymetrix), a total of 870 probe sets were identified to be up or downregulated due to COR treatment under drought stress. Meanwhile, the real-time quantitative PCR (RT-qPCR) method was used to verify some genes; it indicated that there was a good agreement between the microarray data and RT-qPCR results. Our data showed that the differentially expressed genes were involved in stress response, signal transduction, metabolism and tissue structure development. Some important genes response to stress were induced by COR, which may enhance the expression of functional genes implicated in many kinds of metabolism, and play a role in defense response of rice seedling to drought stress. This study will aid in the analysis of the expressed gene induced by COR.

## 1. Introduction

Rice (*Oryza sativa* L.) is widely cultivated as a staple crop throughout the world, and it also is a model monocot plant for molecular and genetic studies, while water deficiency is one of the most important limitations to its growth and grain yield. Shortage of water limits plant growth and crop productivity more than any other single environmental factor [1]. Drought and water deficit can decrease photosynthetic capacity, result in dehydration of plant cells, oxidative damage to chloroplasts, imbalance of photosynthesis and respiration, limit metabolic reactions, and reduce dry matter accumulation and partitioning [2,3]. There are several enzymatic antioxidants to eliminate activated oxygen species, such as catalase (CAT; EC 1.11.1.6), peroxidase (POD; EC 1.11.1.7), superoxide dismutase (SOD; EC 1.15.1.1), ascorbate peroxidase (APX; EC 1.11.1.11) and glutathione reductase (GR; EC 1.6.4.2), and this defence mechanism could be induced during water-deficit stress. Cell membrane stability has been widely used to estimate stress tolerance, and higher membrane stability was correlated with drought stress tolerance [4,5]. Drought stress may particularly trigger the formation of superoxide radical and hydrogen peroxide (H_2_O_2_), which were paralleled by malondialdehyde (MDA) accumulation in rice [6]. MDA, which is a decomposition product of polyunsaturated fatty acids of biomembranes, is an important indicator of membrane damage [7]. The 2,2-dipenyl-1- picrylhydrazyl (DPPH)-ridical scavenging activity is closely correlated with stress tolerance [8], and has been accepted as a measure of radical-scavenging activity [9].

Chemical regulation applying plant growth substance has proved to be potentially beneficial in drought tolerance improving and water-saving agriculture [10]. Coronatine (COR) is a chlorosis-inducing non-host-specific phytotoxin produced by several members of the *Pseudomonas syringae* group of pathovars [11,12,13], and induces a wide array of effects in plants. It is a structural mimic of jasmonates, but more active than jasmonates in some functions, in plants; both share the same receptor, coronatine insensitive1 (COI1) [11,14], so coronatine can manipulate plant hormone signaling to affect defense responses [15,16,17]. It leads to diffuse chlorosis of leaf, anthocyanin production, ethylene (ETH) emission, auxin synthesis, tendril coiling, inhibition of root elongation and hypertrophy [14,18,19]. COR could stimulate the plant metabolism, produce a variety of plant toxins involved in the adjustment of low temperature, salt stress [20,21], water stress [22,23,24,25], and other adversities [26]. Both MeJA and COR can alleviate the adverse effects of drought stress and enhance the ability for water stress resistance through promotion of defense-related metabolism in cauliflower seedlings [27].

Coronatine research has focused on explaining the mechanism of coronatine in hormone pathways [28,29]. Recent research showed that COR interfered with the normal role of salicylic acid (SA) in the interaction with other hormones [30,31], and COR could stimulate the response induced by JA [32]. Biological assays, ultrastructural studies and gene expression studies suggest that COR induces JA biosynthesis, and impacts signaling in tomato via the jasmonic acid, ethylene, and auxin pathways. [14,33]. It is more noticeable that COR has been confirmed to activate the JA responsive genes, with the involvement of some signaling molecules such as hydrogen peroxide, nitric oxide, and Ca^2+^, to induce the production of secondary metabolites such as flavonoids, volatiles, nicotines, and alkaloids [34,35,36]. COR also increased the activities of active oxygen cleavage enzymes, SOD, CAT, POD and GR in two rice cultivars under drought [37]. However, few studies have focused on what the pathways of coronatine are to alleviate water stress in rice at the level of genes.

The objective of this work, therefore, was to evaluate the effects of exogenously applied COR on gene expression and its roles in drought resistance of rice. For these purposes, plants of Nipponbare which have been suffered from drought stress following COR pre-treatment were sampled for Affymetric Gene Chip analysis and RNA isolation. We investigated the changes of expressed genes by Nipponbare gene expression profiles and cDNA microarray analyses, a total of 870 probe sets were identified to be up or downregulated due to COR treatment under drought stress. Some important functional genes response to stress were induced by COR, which play a role in defense response of rice seedling to drought stress. This study is beneficial to clarify the molecular mechanism of COR to alleviate water stress in Rice.

## 2. Results

### 2.1. Growth of Seedlings

As may be seen from the photo of Nipponbare taken at 48 h after 17.5% polyethylene glycol (PEG) treatment (Figure 1), the changes of rice with different treatments were remarkable. Under drought conditions, most of the blade atrophied, and more leaves were rolled into the needle without COR treatment, while COR treatment did not. The data of weight have shown remarkable interaction between COR and PEG. At 7 d after PEG treatment, the dry weight of seedlings was significantly reduced by 37.6% for Nipponbare (Table 1), the fresh mass of rice treated with COR increased 83.7% under drought stress, while COR had no effect on seedling weight before the initiation of drought treatment.

### 2.2. Relative Water Content, Proline Content 

The RWC of rice were decreased under simulated drought stress (Figure 2a). After 72 h exposure to stress, the RWC of leaves (PEG and PEG+COR) were significantly lower than that in non-stressed seedlings (control and COR). However, leaves of the two cultivars under drought stress treated with COR maintained a much higher RWC than the non-treated COR, while, in normal treatments (control and COR), COR had little effect on the RWC. In Nipponbare cultivars, the data of proline content had shown remarkable interaction between COR and PEG (Figure 2b), and the content of proline increased sharply under drought with COR treatment. 

### 2.3. Relative Conductivity and MDA Content

The data of relative conductivity had shown significant changes in rice among different treatments (Figure 2C); PEG-induced drought increased it obviously, but treatment with COR decreased the leakage considerably under drought stress by 21.7%; without drought stress, the relative conductivity was little affected by COR. 

Without drought stress, the MDA content in leaves was more or less similar for control and COR treatments, but the production of MDA increased sharply under drought, while COR pre-treatment reduced the production by 24.3% (Figure 2D).

### 2.4. Superoxide Radical Estimation and DPPH-Radical Scavenging Activity

Compared with the non-stressed plants, the rate of superoxide radical (•O^2−^) in seedlings under drought stress (PEG and PEG +COR) increased significantly (Figure 2E). COR did not affect the production of •O^2−^ in leaves under sufficient water conditions; however, drought stress treatment with COR decreased it by 22.4%.

The DPPH-radical scavenging effect in leaves was decreased by 25.4% due to drought stress, but the decrease was partially alleviated to 14.2% with COR pre-treatment (Figure 2F), while COR had no distinct effect under normal condition.

### 2.5. Functional Analysis of the Transcriptome Map from Rice Nipponbare (Japonica) Treated by COR under Normal and Drought Stress Conditions

In order to investigate what differences in Nipponbare were induced by COR in response to drought stress, sample probe sets were conducted with a microarray analysis using the Affymetrix rice whole genome array. The expression probe in the total chip probe is 38% to 40%, the average was 38.82%, and the present percent value was normal (Table 2). The percent of expressive probe was increased by 0.1% with COR under normal conditions, while the proportion of the probe to COR was decreased by 0.9% under drought stress. It indicated that the response of rice seedlings to COR treatment was different under different growth conditions.

Based on the analysis of expressed genes using the software Cluster3.0, control and COR were clustered together, while PEG and PEG+COR treatments were clustered together. The red represents increased expression, and the deeper of the color stands for the larger of the upregulated expression amount. The green represents downregulated; the deeper the color is, the larger the down expression. There were 870 differentially expressed probes’ responses to COR treatment, and 575 specific responses to COR under drought stress (Figure 3), which accounted for 66.09% of the total number of differentially expressed probes.

According to the probe annotation of Rice Gene Chip, the results of differentially expressed genes on the function of classification using Gene Ontology (GO) classification showed that 49 (occupied 5.6%) probes did not have the corresponding known genes in all 870 differentially expressed probes, which may represent a new mutation in the gene-rich or 3’-end and can not find homologous sequences; the other 821 probes were corresponding to 744 known genes, the functions of genes involve in stress response, signaling, metabolism, photosynthesis, organizational structure development, cell physiological and biochemical reactions, enzyme activity, cell membrane and other aspects (Figure 4).

Kyoto Encyclopedia of Genes and Genomes (KEGG) metabolic pathway analysis showed the function of differentially expressed gene response to COR involved in 20 metabolic pathways under drought stress, including riboflavin metabolism, amino sugar metabolism, alkaloid biosynthesis II, alanine, phenyl-alanyl Acid and aspartic acid metabolism, phenylalanine, tyrosine and tryptophan synthesis, synthesis of diterpene substances, caffeine metabolism, acid metabolism, photosynthesis, porphyrin and chlorophyll metabolism, methane metabolism, pyrimidine metabolism and ribosomes, glutamate metabolism, nicotine and nicotinamide metabolism, carbon fixation, phenyl acid synthesis, and γ-HCH degradation.

According to comprehensive GO classification and KEGG pathway analysis, the differentially expressed genes response to COR under drought stress were mainly related to the protein, carbohydrates, lipids, secondary metabolism and photosynthesis pathway and so on. In addition to the known response genes that were differentially expressed under drought stress, different stress response genes involved in oxidative stress, mechanical damage and biological stress, and other stresses, and genes associated with signal transduction, material transport, ion transport, carbohydrates, fat transfer and others related also responded to COR when exposed to drought (Table 3).

### 2.6. The Differentially Expressed Genes Correlated to JA, ABA and Other Hormone Biosynthetic, and Signaling Pathway under Water Stress

Differentially expressed genes in microarray related to cell elongation, growth and stomatal closure were marked in different signaling pathways (Figure 5). The key elements of pathway were displayed in web-like network under COR treatment, and showed cross talk between JA, ETH, cytokinin (CTK), abscisic acid (ABA) and indole acetic acid (IAA) signaling pathways under drought stress. From our microarray results, *OsWRKY3* (*Os03g0437200*) and *OsWRKY24* (*Os01g0826400*) were upregulated induced by COR, and speculated to influence the pathway of JA and ABA which play crucial roles in multiple aspects of plant drought tolerance. The ETH was abundantly produced though a series of reactions accompanied by 1-aminocyclopropane-1- carboxylate (ACC) synthase (ACS, EC 4.4.1.14) (*Os04g0578000*) which was upregulated to maintain the balance of JA. *OsNAC2* (*Os03g0327800*), the member of the NAC transcription factor family, might be similar to *AtNAC2* and induced by auxin and mediates auxin signaling to promote the growth of plants. Furthermore, MYB/MYC transcription factors were activated by signaling molecules JA and stress response factors, which regulated the expression of downstream genes, in order to reduce the injury caused by stress [38,39,40]. Therefore, the differential expression of response genes may lead to different hormone signaling in addition to enhancing stress tolerance.

### 2.7. Real-Time Quantitative PCR (RT-qPCR) Verifies the Differential Expression of COR Regulated Genes 

To validate the expression results obtained using microarrays, we selected some candidate genes for real-time quantitative PCR (RT-qPCR) analysis to confirm our microarray. They include genes encoding MYB9, Zn-finger, UDP-glucuronosyl, WRKY transcription factor, cytochrome P450 and chalcone biosynthetic related genes. In order to do RT-qPCR validation, the biological samples are the same as what we used in microarray analysis. We designed gene-specific primers from the available full-length cDNAs and performed RT-qPCR (Figure 6). In general, RT-qPCR data are in agreement with the microarray expression data. Additionally, we examined a set of gene expression after drought treatment for the indicated time using RT-qPCR (Figure 7); the fold change of gene expression was maintained at a high level with COR treatment under early drought stress (about within 10 h). ARF2 is a transcriptional suppressor that has been found to be involved in ethylene, auxin, and brassinosteroid pathway to control plant growth and development [41]. MYB is a large family that is involved in abiotic stress tolerance. There were different types such as MYB34, MYB15, MYB111, MYB33, MYB12, MYB4, MYB49, MYB52, MYB59, MYB57, MYB55, MYB70, MYB1, and MYB9 that were upregulated during low temperature stress in *S. spontaneum* [42]. Ethylene, which is interacted with abscisic acid, JA, IAA and SA to regulate stress tolerance, is produced from methionine (Met) via Sadenosylmethionine (AdoMet) by the action of ACC synthase (ACS) and ACC oxidase (ACO) [43]. NAD(P)H dependent 6’-deoxychalcone synthase (DCHS) could be characterized by the cellular activities involved in stress/defense, signal transduction, transport, protein folding, gene regulation, and primary metabolisms, which are critical for plant survival under stress [44]. We suggest that the increased expression of these genes is at least partially a result of enhanced drought tolerance of rice seedlings with COR treatment.

## 3. Discussion

Drought induces a plethora of additional cellular responses that are manifested in changes in whole plant transcriptome and metabolome [45,46,47,48], ultimately resulting in major changes in plant chemical composition. 

### 3.1. Accumulation of Osmotica

Accumulation of osmotica includes ubiquitous chemicals such as sugars and salt ions, and species-specific osmotica such as betaine, glycinebetaine, proline [49]. Proline has been recognized as a multi-functional molecule, accumulating in high concentrations to maintain sustainable growth in response to a variety of content abiotic stresses [50,51]. Pretreatment with COR increased the proline content significantly during drought stress (Figure 2). This is able to protect cells from damage by acting as both an osmotic agent and a radical scavenger. COR alleviated membrane damage by reducing electrolyte leakage and MDA under drought stress (Figure 2). In addition, the levels of superoxide radicals were sharply decreased with the pretreatment of COR under drought (Figure 2). High levels of DPPH-radical scavenging have been correlated with increased stress tolerance [52]. COR increased drought tolerance in rice seedlings through activation of DPPH-radical scavenging effect. As previously reported, COR treatment alleviated salt stress and drought stress by enhancing the activities of CAT, SOD, GPX and GR and the DPPH-radical scavenging capacity [21,37]. We speculated that the inducing effects of COR in water stress tolerance might involve the regulation and expression of functional genes. 

### 3.2. The Significant Expression of Response Genes 

The functions of differentially expressed genes induced by COR under drought stress are involved in stress response, signal transduction, metabolism and other aspects of organizational structure and development [53,54]. Genes encoding zinc finger proteins, ACC synthase, basic leucine zipper protein and calmodulin protein belong to responsive genes. Some drought stress-related zinc finger proteins have been found in rice, tobacco and barley and other plants [55,56]. ACC synthase plays a key role in the catalytic conversion of S-adenosyl-L-methionine (SAM) to ACC, and it is the rate-limiting enzyme of ethylene synthesis [57]. The accumulation of ethylene caused by a significant increase of ACS (Table 3, Figure 5) enhanced the stress response in plants [58,59]. Besides regulating plant growth and development, bZIP (b-zipper) and TFs (transcription factor) remain crucial concerning abiotic stress response such as drought, which is involved in ABA-dependent signal transduction pathways to reduce oxidative damage in Arabidopsis and tobacco under drought and high salt stress [60,61,62]. The calmodulin protein EFA27 also participates in rice dehydration, salt stress response process and the ABA signal transduction pathways [63]. AUX/IAA repressor and ARF transcription activator specifically bind together to regulate the expression of auxin response genes [64]. Under water stress, the significant expression of response genes (Table 3) may together participate in metabolite flux in order to enhance the resistance.

### 3.3. Significantly Upregulated Transcription Factors Associated with Drought Tolerance

TFs were found to play essential roles in multiple abiotic stress responses through binding to cis-regulatory elements in the promoter region of genes to regulating downstream stress-responsive genes [54,65,66]. Thus, genetic modification of the expression of these regulatory genes can greatly influence plant stress tolerance because they further regulate many downstream stress-responsive genes at a given time [67,68]. In our study, there are a total of 18 different types of the transcription factor gene family induced by COR that have been expressed differently under water stress. These genes may play an important part in the process of the drought stress signal transmission and regulation of drought stress response genes, which could be revealed in the presumptive pathway. A *WRKY* gene from a xerophytic evergreen C3 shrub, the creosote bush (*Larrea tridentata*), is an activator of ABA signaling [69]. The expression of WRKY TFs is induced rapidly when the plants are exposed to a variety of stresses signals including salicylic acid (SA) or other molecules [70,71]. WRKY3 regulates expression of JA biosynthesis genes (LOX, AOS, AOC and OPR), thereby increasing the levels of JA and JA-isoleucine, and in turn regulates directly and indirectly drought tolerance, while WRKY24 was found to act as regulators of an ABA-inducible promoter [72,73]. Therefore, significantly upregulated transcription factors (Table 3, Figure 5) induced by coronectin contribute to enhancing drought resistance of rice through ABA and JA levels. 

### 3.4. Changes of the Secondary Metabolic 

The secondary metabolism is the main adaptation mechanism of plants. Terpenoids, alkaloids, phenylpropanoid are the main types of secondary metabolites. Cytochrome, catalyzing the mono-oxidation reaction of many endogenous substrate and exogenous chemicals, is involved in GA deactivation and salt resistance in rice [74]. Cytochrome P450 (CYP) responded to SA significantly improved growth in drought-exposed Triticum aestivum seedlings [75]. Significantly upregulated CYP 450 (Table 3, Figure 5) induced by coronectine contributes to enhancing drought resistance of rice through the SA pathway. 

According to the microarray results and the functional analysis of different gene expression induced by COR in drought conditions, we investigated the tolerant mechanism of rice seedling to drought stress. Some important transcription factors (MYB, WRKY TFs and bZIP TFs) related to stress response are activated or maintained in a high expression level with COR treatment under drought stress, and then may activate expression of genes functioning (calmodulin protein, ACC synthase and CYP 450) in metabolism, such as protein metabolism, carbohydrate metabolism, lipid metabolism, nucleic acid metabolism, photosynthesis and secondary metabolism, hormone content and in maintaining integrity of cell structural and the normal process of physiology. This study provides insights into alleviating water stress of COR in rice (Oryza sativa L.) cultivar nipponbare (Japonica), thus opening applications in the genetic improvement of drought tolerance in rice. 

## 4. Materials and Methods

### 4.1. Plant material, Growth Conditions and COR Treatments

Coronatine was prepared as described in Palmer and Bender (1993) [76]. PEG-6 000 was provided by Sinopharm Chemical Reagent Company. Seeds of Rice (*Oryza sativa* L.) Nipponbare (Japonica) were soaked 36 h in advance disinfection before being sown in the sandbox, then cultured hydroponically in basins containing Kimura B nutrient solution from sandbox when seeds germinated and grew to two leaves in growth chamber characterized by 12 h/12 h photoperiod, 28 °C/25 °C day/night temperature, approximately 60% relative humidity and 400 µ mol m^−2^ s^−1^ photon flux density. At the three leafed stage, 0.1 μmol L-1 COR were added to the solution. Twelve hours after treatment with COR, some of the plants were subjected to water deficit induced by 17.5% PEG-6 000 (−0.44 MPa) in the nutrient solution as the preliminary experiment selected. Each treatment was replicated three times, with three basins containing eighty strains of seedlings, and arranged into a completely randomized design. Ten hours after the drought treatment, plants of Nipponbare were sampled and immediately preserved in liquid nitrogen for Affymetric GeneChip analysis and RNA isolation. Three days after the drought treatment, plants were sampled and preserved by the same method for measurements of MDA level, electrolyte leakage, the activity of DPPH-radical scavenging, proline and relative water content. Seven days later, seedlings were sampled to measure the biomass.

### 4.2. Relative Water Content (RWC), Proline Concentration of Leaves 

Three days after PEG treatment, the leaf RWC was measured on the youngest fully expanded leaves following the method of Sharp et al. [77]. Five plants were examined in each replication. Fresh weight (FW) of leaves was determined immediately after harvest, and then leaf discs were allowed to float on distilled water until fully rehydrated. The discs were weighed for turgid weight (TW). The turgid leaves were dried in a hot air oven at 80 °C to a constant weight and dry weight (DW) was recorded. The RWC of the leaves was calculated as: RWC (%) = (FW − DW)/(TW − DW) × 100. The method for determination of leaf proline content was according to Li [78].

### 4.3. Measurements of Relative Conductivity and MDA Content

Membrane permeability was determined by relative conductivity as described previously in Lutts et al. (1996) [79]. The third fully unfolded leaves were randomly selected from five seedlings and cut in 1 cm segments per replication, and a conductivity meter was used to measure the relative conductivity (Lt/L0). Tissue segments were placed in stoppered vials with 10 mL deionized water and incubated under dark at 25 °C on a rotary shaker (120 rpm). Twenty-four hours later, the electrical conductivity of the bath solution (Lt) was measured. Samples were then autoclaved for 30 min to achieve 100% electrolyte leakage and a final conductivity reading (L0) was recorded upon equilibration at 25 °C. Analysis of electrolyte leakage was performed by relative conductivity of the bath solution before and after autoclaving the tissues.

The level of MDA production was used for estimating the lipid peroxidation. Fresh leaves (0.5 g) were homogenized in 4.0 mL of 10% trichloracetic acid (TCA) and centrifuged at 10,000 rpm for 10 min at 4 °C. The supernatant was assayed for MDA following the method of Sairam and Srivastava (2001) [80].

### 4.4. Superoxide Radical Estimation and DPPH-Radical Scavenging Activity

The superoxide radical (•O2−) was measured by monitoring nitrite formation from hydroxylamine following the method of Verma and Mishra (2005) [81]. The DPPH-radical scavenging activity was determined according to procedures described by Kang and Saltveit (2002) [52].

### 4.5. Affymetrix GeneChip Analysis

Affymetric GeneChip hybridization was performed in CapitalBio Corp (Beijing, China). GeneChip Rice Genome Arrays were used for gene expression profile analysis in this study [82]. Affymetrix^®^ Hybridization Oven 640 (Affymetrix, Inc. West Sacramento, CA, USA), Affymetrix^®^ Fluidics Station 450 and Affymetrix^®^ GeneChip^®^ Scanner 3000 (Affymetrix, Inc., West Sacramento, CA, USA) were used in addition to CEL files that contained estimated probe intensity values, which exported from Affymetrix^®^ GeneChip^®^ Operating Software (Gcos1.4, Affymetrix, Inc. West Sacramento, CA, USA) platform. Software dChip (GDAS, Affymetrix, Inc., West Sacramento, CA, USA) [83,84] was also used to further analysis the results of the data output from Gcos1.4. The annotation of probe sets was obtained from either Affymetrix or the Molecule Annotation System (MAS 3.0) (http://bioinfo.capitalbio.com/mas/) of CapitalBio Corporation.

### 4.6. RNA Isolation and cDNA Preparation

All seedling samples from Nipponbare were homogenized in liquid Nitrogen before isolation of RNA. Total RNA was isolated using TRIZOL^®^ reagent (Invitrogen, CA, USA) and purified using Qiagen RNeasy columns (Qiagen, Hilden, Germany) according to the instructions of the manufacturer. Reverse transcription was performed using Moloney murine leukemia virus (M-MLV; Invitrogen). Briefly, 50 µg of total RNA and 1 µL of oligo d(T) primer (1 pmol) were suspended in a final volume of 10 µL with RNAse-OUT™ (Invitrogen). A master mix (8 µL) containing 5× Superscript RT buffer (Invitrogen), dNTPs (1 µL containing 10 mmol each dNTP), 0.1M DTT (2 µL), and RNA-free ddH2O (1 µL) was prepared and gently added to the RNA-RT mix. Each sample was heated for 3 min at 42 °C, and 2 µL (400 U) Superscript II™ reverse transcriptase (Invitrogen) was then added. The reaction was incubated at 42 °C for 1.5 h, supplemented with 0.5 mol L-1 EDTA (3.5 µL), and then incubated at 65 °C for 10 min to stop the reaction and hydrolyze template RNA. 

### 4.7. Real-Time Quantitative PCR (RT-qPCR) Analysis

Three biologically independent replicates of RT-qPCR were conducted on 10 genes. Each replicate used aliquots of the same RNA samples used for microarrays. All gene-specific primers were designed by Primer 3 (http://frodo.wi.mit.edu/primer3/input.htm) using the full-length Oryza sativa L. cDNAs available from GenBank (Appendix A), and the amplification of actin gene (Genbank: NM-197297) was used as an internal control to normalize all data. Real-time quantitative RT-PCR was performed on a 7500 real-time PCR system (Applied Biosystems, Foster, CA, USA) using SYBR^®^ Premix Ex Taq ™ (Perfect Real Time) (TaKaRa Code: DRR041A, Dalian, China). According to the manufacturer’s protocol, 1.5 µL cDNA, 0.4 µL PCR forward/reverse primer (10 µmol), 10 µL 2× SYBR^®^ Premix Ex Taq™ and 0.4 µL ROX Reference Dye II (50×) were suspended in a final volume of 20 µL with ddH2O. RT-qPCR cycling conditions consisted of an initial polymerase activation step at 95 °C for 30 sec, 40 cycles of 5 sec at 95 °C, and 35 sec at 60 °C. Melt-curve analysis was performed to monitor primer–dimer formation and the amplification of gene specific products. The relative quantification method (DDCT) was used to evaluate quantitative variation between replicates examined.

### 4.8. Statistical Analysis

The experiment was repeated three times and, since no interactions occurred among repetitions, the results were pooled for analysis of variance following the general linear model of the Statistical Analysis System (SAS) (SAS Institute Inc., Cary, NC, USA). Treatment means were compared using the least significant difference test at the 5% level of probability.

## 5. Perspectives

Previous effects have showed that many drought-responsive genes, identified by RNA-seq and proteomics, have had their roles in alleviating water stress during water deficiency functionally confirmed. However, our understanding of gene expression profiles deciphering the pathways of Coronatine alleviating water stress in rice is just beginning. First, there are still many unresolved questions regarding the activity of COR in modulating phytohormone pathways. The main pathway of coronatine to alleviate water stress in rice needs to be explored more. Second, different drought-response genes have been identified, while their expression pattern, cellular or tissue localization, interactions and functions have not yet been reported. The function of most suppressed genes induced by COR remains unclear in drought conditions. Further analysis of COR-responsive genes and the potential receptors for COR needs to be investigated, which will help elucidate the mechanism of action for COR. Furthermore, the use of a gene editing approach to edit these functional genes for molecular breeding would also be beneficial.

## Figures and Tables

**Figure 1 ijms-20-02543-f001:**
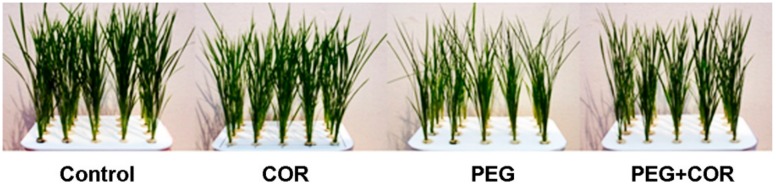
Effect of cornatine on Nipponbare rice seedlings under drought at 48 h. Control, Normal conditions without COR and PEG treatment; PEG, Water deficit treatment induced by PEG for 10 h without COR pretreatment; COR+PEG, Pretreatment with 0.1 μM COR for 12 h before PEG treatment; COR, Treatment with 0.1 μM COR alone, without PEG treatment; the same for the following tables and figures.

**Figure 2 ijms-20-02543-f002:**
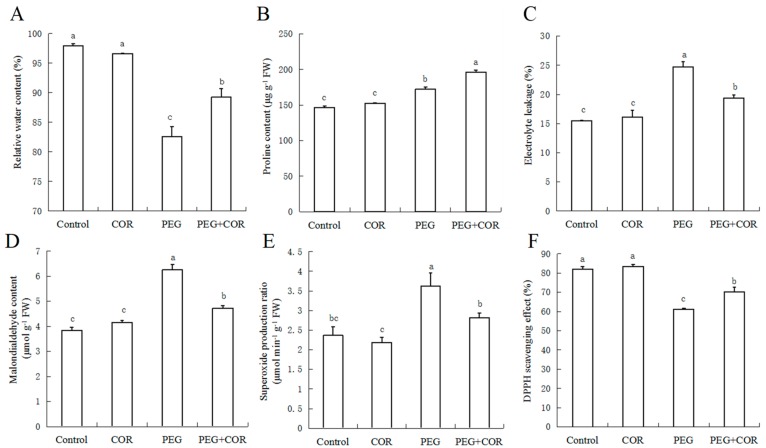
Effects of cornatine on relative water content (**A**), proline content (**B**), electrolyte leakage (**C**), MDA content (**D**), superoxide production ratio (**E**) and DPPH scavenging activity (**F**) in leaves of rice seedlings under drought condition at 3 d. Values are means of four experiments each with three replications; bars with different letters are significantly different (*p* < 0.05). FW, fresh weight.

**Figure 3 ijms-20-02543-f003:**
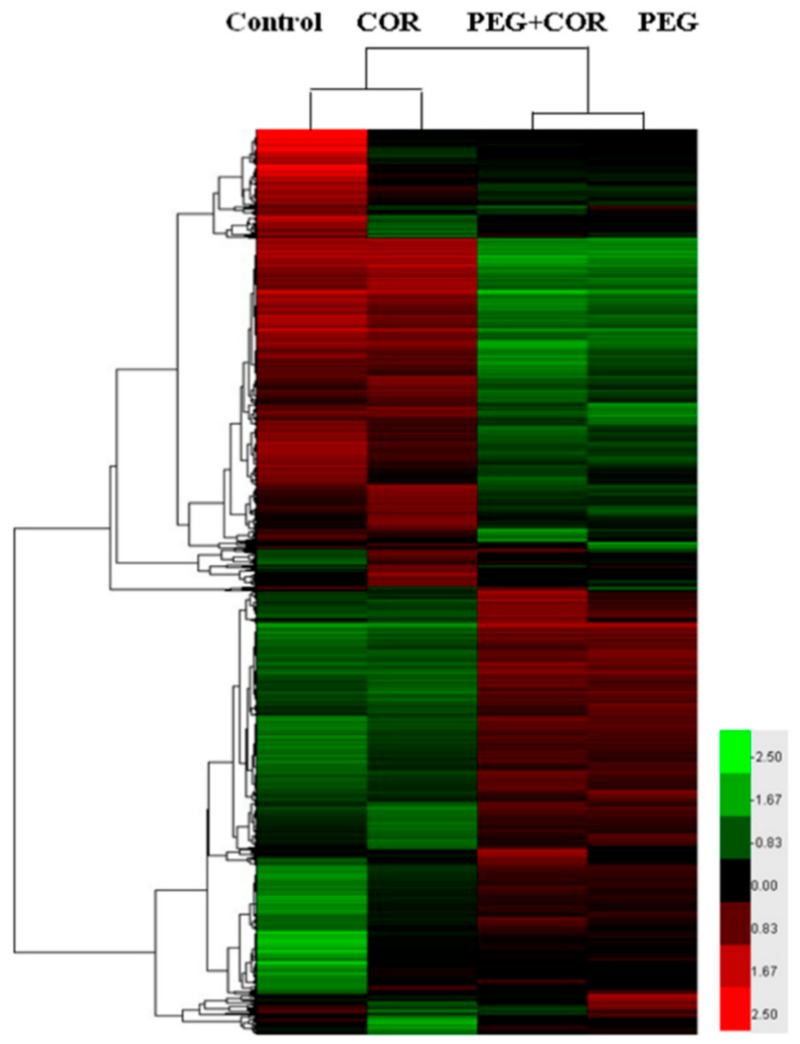
Effects of coronatine on the transcriptional profiles of rice leaf blades under drought stress. Cluster analysis of differential display genes response to COR under drought stress condition. Each column represents Log2 ratio (Cy5/Cy3) of the different gene. The black color represents the ratio of 1:1; the red is for greater than 1:1.

**Figure 4 ijms-20-02543-f004:**
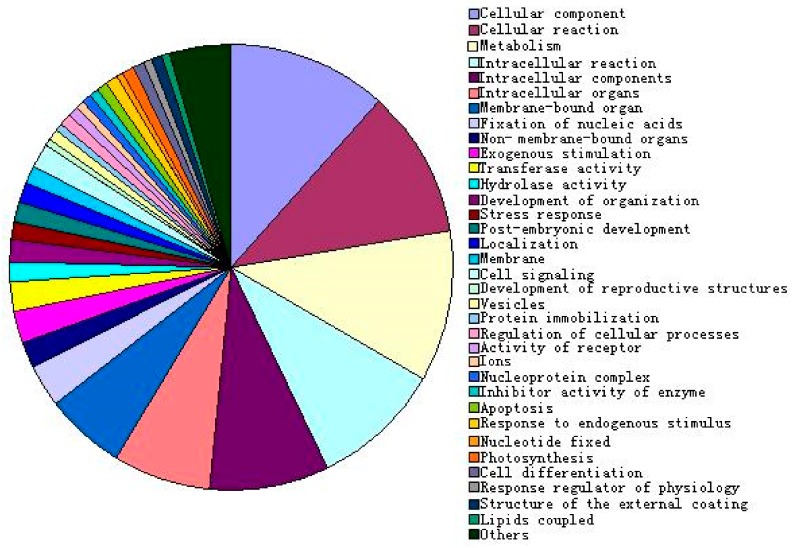
Function analysis of differential display genes response to cornatine under drought stress.

**Figure 5 ijms-20-02543-f005:**
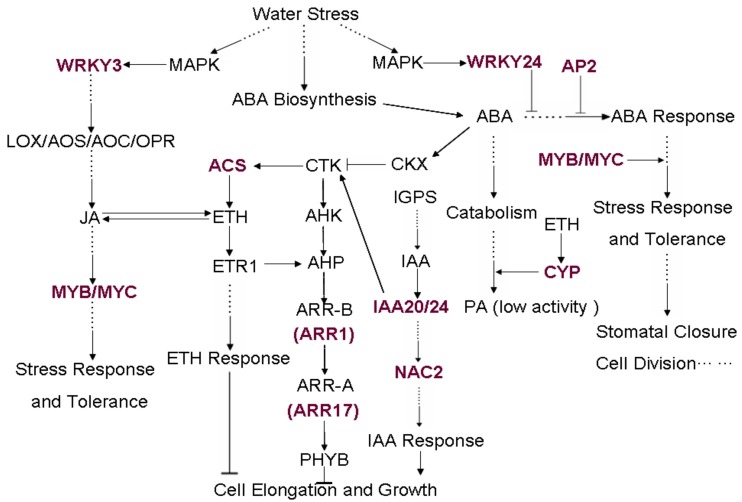
Gene net correlated to JA, ABA and other hormone biosynthetic, signaling and response pathway under water stress. The key elements of pathway are displayed in web-like network and are connected by arrows. A previously established link among IAA, ABA, Ethylene, CTK and cell elongation and growth is indicated. The red and overstriking words represent differentially expressed genes induced by COR in microarray. Arrows and t-bars represent positive and negative effects, respectively. Solid lines indicate effects that occur through direct interaction, whereas dotted lines indicate effects that have not yet been shown to occur through direct interaction.

**Figure 6 ijms-20-02543-f006:**
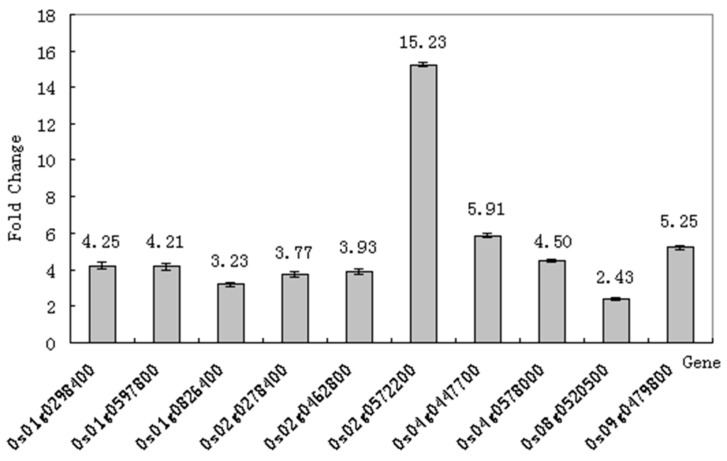
Transcriptional changes were analyzed using RT-qPCR. The expression gene annotation can be found in Table 3, the primers used for the RT-qPCR are presented in Appendix A, while the fold change values represent the average of three independent biological replicates for comparison with microarray data, the actin gene was used as an internal control. Values are means of three experiments each with three replications; PEG, Water deficit treatment induced by PEG for 10 h without COR pretreatment; COR+PEG, Pretreatment with 0.1 μM COR for 12 h before PEG treatment. The sample was collected 10 h after treatment.

**Figure 7 ijms-20-02543-f007:**
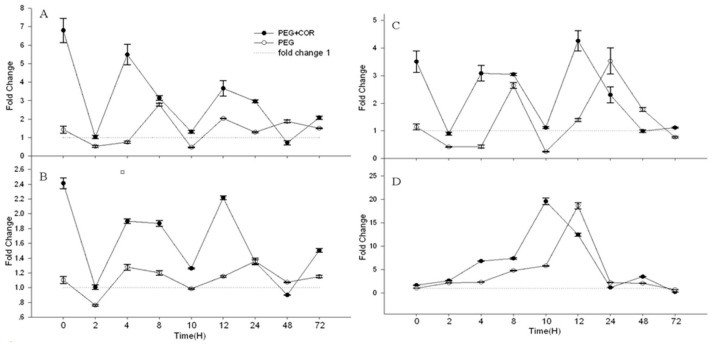
The changes of four genes expression in relation with the time. The expression data are presented for (**A**): Os01g0298400 (*MYB9*), (**B**): Os08g0520500 (*ARF2*), (**C**): Os04g0578000 (*ACS*) and (**D**): Os04g0447700 (*DCHS*). PEG, Water deficit treatment induced by PEG for 10 h without COR pretreatment; COR+PEG, Pretreatment with 0.1 μM COR for 12 h before PEG treatment. The sample was collected in 2 h, 4 h, 8 h, 10 h, 12 h, 24 h, 48 h, and 72 h.

**Table 1 ijms-20-02543-t001:** Effects of coronatine on biomass of Nipponbare rice seedlings under drought stress at seven days after the treatment.

Treatment	Fresh Weight (g/plant)	Dry Weight (g/plant)
Control	0.727a	0.101a
COR	0.700a	0.098a
PEG	0.283c	0.063c
PEG+COR	0.520b	0.073b

Values are means of four experiments each with three replications. Means within columns bearing the same letters are not significantly different (*p* < 0.05).

**Table 2 ijms-20-02543-t002:** Percent of present probe in four treatments.

Treatment	Total	Present	P Call (%)
Control	57381	21806	38.00
PEG	57381	22827	39.78
COR	57381	21827	38.04
PEG+COR	57381	22632	39.44

**Table 3 ijms-20-02543-t003:** Different expressing genes related to stress, transcription factor and others response to coronatine under drought stress.

Catalogue	Gene Name	Fold Change	Annotation
Response to Stress	Os02g0572200	12.43	Zn-finger, RING domain containing protein
	Os06g0166500	4.98	AUX/IAA protein family protein
	Os09g0474000	4.18	Basic-leucine zipper (bZIP)
	Os07g0686800	3.36	Serine/threonine protein kinase
	Os09g0255400	3.06	Indole-3-glycerol phosphate synthase (IGPS)
	Os04g0578000	3.08	ACC synthase
	Os06g0549900	2.71	FAD linked oxidase, N-terminal domain containing protein
	Os04g0511200	2.26	EFA27 for EF hand, abscisic acid
	Os09g0522000	0.49	CBF-like protein
	Os08g0474000	0.42	AP2 domain containing protein RAP2.6
	Os01g0580500	0.25	1-aminocyclopropane-1-carboxylate oxidase
	Os12g0139400	0.09	Two-component response regulator ARR17
Transcription Factor Gene	Os08g0520500	22.63	Auxin response factor 2
	Os01g0298400	3.67	MYB9
	Os09g0417800	3.31	DNA-binding WRKY domain containing protein
	Os03g0327800	3.19	NAC-domain containing protein 29 (NAC2)
	Os01g0826400	2.82	WRKY transcription factor 24
	Os01g0904700	2.79	Two-component response regulator ARR1
	Os03g0437200	2.52	Transcription factor WRKY3
	Os04g0605100	2.20	WRKY transcription factor 68
	Os02g0462800	2.18	WRKY transcription factor 42
Correlate to Photosynthesis and Others	Os01g0597800	9.34	UDP-glucuronosyl/UDP-glucosyltransferase family protein
	Os03g0277700	9.28	Protein of unknown function DUF26 domain
	Os04g0447700	4.48	NAD(P)H dependent 6’-deoxychalcone synthase
	Os06g0569500	3.40	Ent-kaurene oxidase (AtKO1) ( Cytochrome P450 701A3)
	Os06g0549900	2.71	FAD linked oxidase, N-terminal domain containing protein
	Os03g0709300	2.34	Plastocyanin-like domain containing protein
	Os02g0194700	2.26	Plant lipoxygenase family protein
	Os01g0600900	0.50	Chlorophyll a-b binding protein 2, chloroplast precursor
	Os12g0292400	0.49	Ribulose 1,5-bisphosphate carboxylase
	Os03g0280000	0.43	ABC transporter protein
	Os07g0675400	0.16	Aminoacyl-tRNA synthetase

The multiple is behalf of gene fold change in the table, make the standard of more than 2 times (the absolute value of Log2 ≥ 1) determine the differentially expressed genes, the value > 2 represents up-regulated expression, otherwise it is down-regulated expression.

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
