# Peer review of "Gene Expression Profiles Deciphering the Pathways of Coronatine Alleviating Water Stress in Rice (Oryza sativa L.) Cultivar Nipponbare (Japonica)"

_ijms, 2019, doi:10.3390/ijms20102543_

Round 1

Reviewer 1 Report

The manuscript entitled ''Gene Expression Profiles Deciphering the Pathways of Coronatine 
Alleviating Water Stress in Rice (Oryza sativa L.) Cultivar Nipponbare (Japonica)" studied the effects of COR on the drought stress of rice (Oryza sativa L.). 

The authors addressed my concerns and carried out the comments I recommended properly. 

However, I still suggest that the authors do the following minor revisions;

1. The introduction can still be improved, including such information highlighting the objectives of this current study.

2. The discussion can still be improved, including the reasons of getting such results and potential roles of the treatments used.

3. The authors should also revise the conclusions sections and include all the significant results and conclusions in this section so that the upcoming researches can build on this current work.

Author Response

1.      The introduction can still be improved, including such information highlighting the objectives of this current study.

Response: We thank reviewer raising this point. Now we added the objective of this current study, which has been marked in red color.

2.      The discussion can still be improved, including the reasons of getting such results and potential roles of the treatments used.

Response: We sincerely apologized for this. We have modified the discussion in the manuscript, which has been marked in red color.

3.      The authors should also revise the conclusions sections and include all the significant results and conclusions in this section so that the upcoming researches can build on this current work.

Response: We thank reviewer raising this point. Now we added the perspective, which has been marked in red color.

Reviewer 2 Report

I am sorry but still the MS contains many inaccuracies and errors so that it deserve a major revision.

In fact:

In Abstract authors stated that “Pre-treatment with COR significantly increased the biomass” but that is not the case from Table 1 where COR+PEG values are lower than PEG.

Figure 1 doesn’t show any difference between PEG+COR and PEG;

Table 1. the order of the samples is different from Fig.1 and Fig. 2 (a series Control – COR – PEG – PEG + COR), and PEG+COR is the correct definition.

Between lines 104 and 105  a “72 h” exposure to stress is indicated, but in Fig. 1 the time is 48 h, for table 1 it is not indicated (maybe 10 d as stated in M&M line 373) and for Figure 2 is 3 days.

The legend of fig. 2 is completely wrong as there are no correspondence between description and the six series of histograms (proline content is not Fig. 2b but Fig. 2d).

Figure 3b needs to be explained as the description and numbers in the text (lines 140-142) doesn’t correspond to the Venn diagrams; also, it is meaningless a comparison Control vs. PEG because  before the Authors have mainly compared PEG vs. PEG+ COR.

Figure 6 and 7 are of low quality.

In M&M the ages of the different samples are no clearly indicate.

A clear Conclusion is missing.

Author Response

1.In Abstract authors stated that “Pre-treatment with COR significantly increased the biomass” but that is not the case from Table 1 where COR+PEG values are lower than PEG.

Response: We sincerely apologized for this. Due to our oversight, the processing name does not match the biomass. We have revised the name of the treatment to finally show that COR pretreatment significantly improves the biomass, which has been marked in red color.

2. Figure 1 doesn’t show any difference between PEG+COR and PEG;

Table 1, the order of the samples is different from Fig.1 and Fig. 2 (a series Control – COR – PEG – PEG + COR), and PEG+COR is the correct definition.

Response: We sincerely apologized for this. We have modified the definition and order mistakes in the manuscript, which have been marked in red color.

3. Between lines 104 and 105 a “72 h” exposure to stress is indicated, but in Fig. 1 the time is 48 h, for table 1 it is not indicated (maybe 10 d as stated in M&M line 373) and for Figure 2 is 3 days.

The legend of fig. 2 is completely wrong as there are no correspondence between description and the six series of histograms (proline content is not Fig. 2b but Fig. 2d).

Response: We sincerely apologized for this. We have modified the definition and order mistakes in the manuscript, which have been marked in red color.

4.Figure 3b needs to be explained as the description and numbers in the text (lines 140-142) doesn’t correspond to the Venn diagrams; also, it is meaningless a comparison Control vs. PEG because before the Authors have mainly compared PEG vs. PEG+ COR.

Response:

5.Figure 6 and 7 are of low quality.

Response: We thank reviewer raising this point. Now we replaced those figures with ones with high resolution.

6.In M&M the ages of the different samples are no clearly indicate.

Response: we revised the Method part.

7.A clear Conclusion is missing.

Response: We sincerely apologized for this. We have modified the discussion in the manuscript, which has been marked in red color.

Reviewer 3 Report

Author studied the effects of Coronatine (COR) on the drought stress of rice  (Oryza sativa L.). Pre-treatment with COR significantly increased the biomass, relative water and proline content, and DPPH-radical scavenging activity, decreased the electrolyte leakage and MDA content in order to maintain the stability of cell membrane. They determined how COR alleviates water stress by Nipponbare gene expression profiles and cDNA microarray analyses. They showed differentially expressed genes were involved in stress response, signal transduction, metabolism and tissue structure development. Some important genes response to stress were induced by COR, which may enhance the expression of functional genes implicated in5 many kinds of metabolism, and play a role in defense response of rice seedling to drought stress. This study will aid in the analysis of the expressed gene induced by COR.

·         The topic is within the scope of this journal.  However, the manuscript preparation does not reach to the standards of scientific publication.

·         In introduction and discussion part author needs to cite more recent references.

·         Author need to analyze the Chlorophyll and anthocyanin content of treated and control samples.

·         Author need to provide the details Root length and morphology of treated and control samples.

·         English of the MS needs to be greatly improved. The English of the whole article has to be checked carefully to eliminate linguistic errors.

·         Authors should add some more details in materials and methods.

·       In Conclusion, authors should add significance of this research to potential practical application.

Author Response

1. In introduction and discussion part author needs to cite more recent references.

Response: Thank you for your valuable comments. We have added relevant recent references in introduction and discussion and marked them in red.

2. Author need to analyze the Chlorophyll and anthocyanin content of treated and control samples.

Response: Many thanks to the reviewer for pointing out this deficiency. We have modified the article according to your requirements and marked it with red words.

3. Author need to provide the details Root length and morphology of treated and control samples.

Response: We sincerely thank the reviewers for their valuable Suggestions. We have added the related data of rice root into the paper, which was marked in red color.

4. English of the MS needs to be greatly improved. The English of the whole article has to be checked carefully to eliminate linguistic errors.

Response: We sincerely apologized for this. After reading and checking the full text carefully, we have found and corrected the errors in these language expressions, and marked them in red font.

5. Authors should add some more details in materials and methods.

Response: We thank reviewer giving us the suggestion. We have added more details and materials and methods as you said, which was marked in red color.

6. In Conclusion, authors should add significance of this research to potential practical application.

Response: Many thanks to the reviewer for pointing out this deficiency. We have added relevant content to the article to illustrate significance of this research to potential practical application, which were marked in red color.

Round 2

Reviewer 2 Report

The MS was really improved but stile some few points deserve e revision:

-        Table1: the legend is not well written, and it lacks the timing (7 days after the treatment);

-        Figure 3b: it is again very difficult to understand and again the numbers indicated in the text (lines 142 and 143) do not correspond to numbers in Figure 3b; moreover, following the figure legend the Venn diagrams should represent the genes regulated by COR comparing COR vs. Control and PEG+COR vs. PEG;

-        Figure 6 and Figure 7: the indications of the age of the samples and the type of sample (COR or PEG+COR or whatever?) are missing;

-        Table 4 should be Supplementary material;

-        The format of References should be checked.

Author Response

1.      Table1: the legend is not well written, and it lacks the timing (7 days after the treatment);

Response: We thank reviewer raising this point. We have added it.

2.      Figure 3b: it is again very difficult to understand and again the numbers indicated in the text (lines 142 and 143) do not correspond to numbers in Figure 3b; moreover, following the figure legend the Venn diagrams should represent the genes regulated by COR comparing COR vs. Control and PEG+COR vs. PEG;

Response: We thank reviewer raising this point. It really make people confused. Actually, It is based on the same factor COR, and compare the difference between Control and PEG to tell people that COR play a critical role under drought stress conditions. Anyway we delete it in case it confused people.

3.      Figure 6 and Figure 7: the indications of the age of the samples and the type of sample (COR or PEG+COR or whatever?) are missing;

Response: We thank reviewer raising this point. We have added it.

4.       Table 4 should be Supplementary material;

Response: We thank reviewer raising this point. We put it in supplementary material.

5.       The format of References should be checked.

Response:  We sincerely apologized for this. We revised them.

Reviewer 3 Report

requested correction was carried out. 

Author Response

Thanks for your confirmation.

This manuscript is a resubmission of an earlier submission. The following is a list of the peer review reports and author responses from that submission.

Round 1

Reviewer 1 Report

Review Comments

In the manuscript entitled ‘’Gene Expression Profiles Deciphering the Pathways of Coronatine Alleviating Water Stress in Rice (Oryza sativa L.) Cultivar Nipponbare (Japonica)”, the authors studied the effects of COR on the droutht stress of rice (Oryza sativa L.). Pre-treatment with COR significantly increased the biomass, relative water and proline content, and DPPH-radical scavenging activity, decreased the electrolyte leakage and MDA content in order to maintain the stability of cell membrane. Meanwhile, the authors determined how COR alleviates water stress by Nipponbare gene expression profiles and cDNA microarray analyses. The data showed the differentially expressed genes were involved in stress response, signal transduction, metabolism and tissue structure development. Some important genes response to stress were induced by COR, which may enhance the expression of functional genes implicated in many kinds of metabolism, and play a role in defense response of rice seedling to drought stress. The manuscript is well written. However, the following revisions should be carried out;

- Abstract

Main findings should be included in the abstract.

English should be corrected in the abstract (i.e. the word “but” should be added in line 4 in the abstract.

- Introduction

The introduction is clear, well explained and covered the objectives, however recent references should be provided.

- Results

The visibility of Figures 2 and 4 is very low, so please replace those figures with ones with high resolution.

Please explain the section of real-time PCR in more details; i.e. what should such genes assist in the defence strategy? etc..

- Discussion

The discussion is somewhat well-presented but is still poor and should be improved.

The current data should be linked with the previous findings and reasons should be given and discussed if differences found.

- Methods

Different sections should include more details on how you prepare the samples to take such measurements.

-References

Recent references on recently published rice water stress findings should be included.

Reviewer 2 Report

The MS presents a lot of data, but in a confused way, concerning  the effect of water stress (PEG) and coronatine. At least the two treatments need to be separately discussed and analyzed. Anyway, it seems that no  interesting genes are activated / repressed following the COR pre-treatment.

In addition:

-        The English must be revised (e.g coronatine or cornatine?);

-        The phrase “Pre-treatment with COR significantly increased the biomass, relative water and proline content” (Abstract lines 19-20) does not fit with Figure 1 and Table 1 where both dry weight and fresh weight are at minimum values;

-        Line 104: “leaves of the two cultivars” – which is the second cultivar?

-        Figure 1: 48 h from what?

-        Table 1: plants at 48 H or 3 d?

-        Figure 2: data must b expressed for dry weight;

-        The sequence Control, COR, PEG, COR+PEG must be kept constant in Figures and Tables;

-        Figure 3: A Venn diagram including a comparison between PEG e COR+PEG is appropriate;

-        Figure 6: please add the time after treatments;

-        Figure 7: what is the time zero?

Reviewer 3 Report

In this study, the authors investigated the effects of on the drought stress in rice (Oryza sativa L.) Cultivar Nipponbare (Japonica). Coronatine (COR) is a structural and functional analog of methyl jasmonic acid (MeJA) and can alleviate stress on plant. Pre-treatment with COR significantly increased the biomass, relative water and proline content, and DPPH-radical scavenging activity, decreased the electrolyte leakage and MDA content in order to maintain the stability of cell membrane. They determined how COR alleviates water stress by Nipponbare gene expression profiles and cDNA microarray analyses. The real-time quantitative PCR (RT-qPCR) method was used to verify some genes; it indicated that there was a good agreement between the microarray data and RT-qPCR results. They results concluded the differentially expressed genes were involved in stress response, signal transduction, metabolism and tissue structure development.

The topic is within the scope of this journal.  However, the manuscript preparation does not reach to the standards of scientific publication.

English of the MS needs to be greatly improved. The English of the whole article has to be checked carefully to eliminate linguistic errors.

In introduction and discussion part author needs to cite more recent references.

Author need to analyze the effects  on the drought stress in chlorophyll, anthocyanin and H2O2 content of rice plants (treatment and control).

Discussion need to be improved.